# Structural Insights into Retinal Guanylate Cyclase Activator Proteins (GCAPs)

**DOI:** 10.3390/ijms22168731

**Published:** 2021-08-13

**Authors:** James B. Ames

**Affiliations:** Department of Chemistry, University of California, Davis, CA 95616, USA; jbames@ucdavis.edu; Tel.: +1-1530-752-6358

**Keywords:** phototransduction, retinal guanylate cyclase, calcium, GCAP1, GCAP2, GCAP5

## Abstract

Retinal guanylate cyclases (RetGCs) promote the Ca^2+^-dependent synthesis of cGMP that coordinates the recovery phase of visual phototransduction in retinal rods and cones. The Ca^2+^-sensitive activation of RetGCs is controlled by a family of photoreceptor Ca^2+^ binding proteins known as guanylate cyclase activator proteins (GCAPs). The Mg^2+^-bound/Ca^2+^-free GCAPs bind to RetGCs and activate cGMP synthesis (cyclase activity) at low cytosolic Ca^2+^ levels in light-activated photoreceptors. By contrast, Ca^2+^-bound GCAPs bind to RetGCs and inactivate cyclase activity at high cytosolic Ca^2+^ levels found in dark-adapted photoreceptors. Mutations in both RetGCs and GCAPs that disrupt the Ca^2+^-dependent cyclase activity are genetically linked to various retinal diseases known as cone-rod dystrophies. In this review, I will provide an overview of the known atomic-level structures of various GCAP proteins to understand how protein dimerization and Ca^2+^-dependent conformational changes in GCAPs control the cyclase activity of RetGCs. This review will also summarize recent structural studies on a GCAP homolog from zebrafish (GCAP5) that binds to Fe^2+^ and may serve as a Fe^2+^ sensor in photoreceptors. The GCAP structures reveal an exposed hydrophobic surface that controls both GCAP1 dimerization and RetGC binding. This exposed site could be targeted by therapeutics designed to inhibit the GCAP1 disease mutants, which may serve to mitigate the onset of retinal cone-rod dystrophies.

## 1. Introduction

### 1.1. Ca^2+^-Sensitive Regulation of RetGC Coordinates Visual Recovery

Visual excitation of retinal rod and cone photoreceptors is triggered by a phototransduction cascade in which light excitation activates a photoreceptor-specific phosphodiesterase that in turn hydrolyzes cGMP (see reviews by [1,2]). The light-induced lowering of cGMP levels in photoreceptor cells causes the closure of cGMP-gated cation channels in the plasma membrane, resulting in membrane hyperpolarization (see reviews by [3,4]). The light-induced membrane hyperpolarization rapidly recovers back to the resting potential of the dark state when the light stimulus is removed in a process known as visual recovery. The recovery phase of phototransduction involves replenishing the photoreceptor cGMP levels [5] by the Ca^2+^ sensitive activation [6,7] of retina-specific guanylate cyclases (RetGCs) [8,9]. The Ca^2+^-dependent activity of RetGC is controlled by intracellular domains [10,11] that interact with soluble EF-hand Ca^2+^ sensor proteins, called guanylate cyclase activator proteins (GCAP1-5, see Figure 1) [8,12,13,14,15,16].

Light-induced closure of cGMP-gated channels in vertebrate rod and cone photoreceptors causes a 10-fold decrease in the cytosolic free Ca^2+^ concentration [17,18]. RetGC catalysis is activated by Ca^2+^-free GCAPs in light-activated photoreceptors [8,12,13,19,20], whereas the cyclase activity is inhibited by Ca^2+^-bound GCAPs in dark-adapted photoreceptors [5,19,21]. During visual recovery, a photoreceptor cell exhibits a more than 10-fold increase in cGMP production due to the Ca^2+^-sensitive activation of RetGC by GCAPs [5,22] and is a critical step for controlling the recovery rate of a single-photon response [4,5] as well as the cone response to stronger light stimuli [23].

### 1.2. Ca^2+^/Mg^2+^ Binding to GCAPs Control Activation of RetGC

GCAP proteins bind to and activate RetGC in light-activated photoreceptors that contain low Ca^2+^ levels (less than 50 nM) and physiological Mg^2+^ levels (1 mM) [24,25,26,27]. Thus, GCAP proteins that exist in light-activated photoreceptors activate RetGC and are called the activator state. At least one Mg^2+^ binds to GCAP1 in the activator state [26], and NMR studies reveal that Mg^2+^ is bound to GCAP1 at the second EF-hand (EF2 in Figure 1) [28]. The apo-state of GCAP1 (Ca^2+^-free/Mg^2+^-free) contains a regular secondary structure [29] but does not adopt a stable three-dimensional fold [25,28]. The Ca^2+^-free/Mg^2+^-free GCAPs form a flexible molten-globule state, which could explain why GCAPs do not activate RetGC in the absence of Mg^2+^ [24]. Thus, Mg^2+^ binding to GCAP1 is required to stabilize its protein structure to promote activation of RetGC [8,25,30]. By contrast, Ca^2+^ binding to GCAP1 (in place of Mg^2+^ binding) stabilizes a distinct structure important for the inhibition of RetGC [21]. Ca^2+^ binds to GCAPs at the second, third, and fourth EF-hands (EF2, EF3, and EF4 in Figure 1) [31,32]. The apparent dissociation constant for Ca^2+^ binding to GCAPs is 100 nM [24,28], whereas Mg^2+^ binds to GCAPs in the micromolar range [25,27,28]. Dark-adapted rod cells have relatively high cytosolic Ca^2+^ levels ([Ca^2+^]_free_ = 250–500 nM [18], which implies that GCAPs are nearly saturated with Ca^2+^ in dark-adapted rod cells. Light-activation of the rod cell causes a dramatic lowering of the cytosolic Ca^2+^ level ([Ca^2+^]_free_ = 5–50 nM [17,18,33]) while the Mg^2+^ level remains fixed at [Mg^2+^]_free_ ~ 1 mM [34]. Therefore, in light-adapted rods, GCAPs are bound to Mg^2+^ instead of Ca^2+^. In essence, the Mg^2+^-bound/Ca^2+^-free GCAPs in light-activated photoreceptors turn on the synthesis of cGMP to help restore the dark-adapted photoreceptor during visual recovery [8,12,13], whereas Ca^2+^-saturated GCAPs turn off the synthesis of cGMP in the resting dark state [19,21].

### 1.3. Mutations in GCAP1 Cause Retinal Disease

Mutations in GCAP1 that weaken or disable Ca^2+^ binding to the EF-hands cause GCAP1 to constitutively activate RetGC in rod and cones. Some of these mutations (Y99C, D100G, E111V, and E155G) are genetically linked to retinal diseases known as cone-rod dystrophies [29,35,36]. For example, the GCAP1 mutants (Y99C [21,37], D100G [38], E111V [39] and E155G [40,41]) each prevent Ca^2+^ binding to EF3 or EF4 under physiological conditions, which enables the Ca^2+^-free/Mg^2+^-bound GCAP1 activator state to persist in both light-activated and dark-adapted photoreceptors. In essence, these constitutively active GCAP1 mutants fail to turn off the cyclase activity in dark-adapted photoreceptors and cause persistent activation of RetGC [42,43]. This constitutive activation of RetGC causes elevated cGMP levels in photoreceptor cells that promote apoptosis and disease [42,44,45]. Future studies are needed to discover therapeutic agents that bind specifically to the constitutively active mutants of GCAP1 (Y99C, D100G, E111V, and E155G) to block or prevent their constitutive activation of RetGCs, which may diminish or slow down the onset of cone-rod dystrophies.

## 2. Results and Discussion

### 2.1. Structural Architecture of GCAPs

Mammalian photoreceptors have two different GCAP isoforms (GCAP1 and GCAP2 in Figure 1) that are more than 65% identical to GCAP homologs found in zebrafish photoreceptors (GCAP3-5 in Figure 1). All of the GCAPs contain ~200 residues, 4 EF-hand motifs (highlighted in color in Figure 1), a myristoyl group covalently attached to the N-terminal glycine, and non-conserved residues at the N- and C-termini (α1 and α11 in Figure 1). The second, third, and fourth EF-hands each bind to Ca^2+^ or Mg^2+^ as described above. The first EF-hand (EF1) does not bind to Ca^2+^ or Mg^2+^ because of unfavorable residues in the EF-hand binding loop (Cys29 in GCAP1 or Arg25 in GCAP3, see Figure 1). The lack of metal binding to the first EF-hand allows it to adopt an unusual structure that interacts with the N-terminal myristoyl group [32,46,47]. Outside of the core EF-hand region, the non-conserved helices (α1 and α11, highlighted purple in Figure 1) both form contacts with the myristoyl group [32]. Atomic-level structures are known for Ca^2+^-bound forms of GCAP1 [32] and GCAP2 [31], and Mg^2+^-bound/Ca^2+^-free GCAP1 [48] as described below.

#### 2.1.1. NMR Structure of GCAP2

The NMR structure of the Ca^2+^-saturated and unmyristoylated GCAP2 (Figure 2A) was the first atomic-resolution structure of a GCAP protein [31]. The first 20 amino acids from the N-terminus and the last 19 residues from the C-terminus in unmyristoylated GCAP2 could not be resolved by NMR (see dotted lines in Figure 2A). The core region of GCAP2 (residues 23–185) contains 4 EF-hands that are structurally similar to the EF-hands in Ca^2+^-bound recoverin [49,50]. An important structural difference is that Ca^2+^ is bound at EF2, EF3, and EF4 in GCAP2, in contrast to recoverin where Ca^2+^ is bound only at EF2 and EF3 [51]. The lack of N-terminal myristoylation in the NMR structure of GCAP2 may contribute to the structural disorder at the N- and C-termini (dotted lines in Figure 2A). This could account for why the N-terminal myristoyl group in GCAP2 is exposed to the exterior in the presence of lipid bilayer membranes [52,53], which could enable the myristoyl group to anchor GCAP2 to membranes [54,55]. The exposed and unstructured N-terminal region in the GCAP2 NMR structure may explain why GCAP2 can exhibit Ca^2+^-dependent membrane binding, whereas GCAP1 does not [56].

#### 2.1.2. Crystal Structure of GCAP1

The x-ray crystal structure of myristoylated GCAP1 (Figure 2B) showed the N-terminal myristoyl group to be sequestered inside the protein [32]. The four EF-hands in GCAP1 (Figure 1 and Figure 2B) are grouped into two globular domains: the N-domain is comprised of EF1 and EF2 and the C-domain is comprised of EF3 and EF4. Ca^2+^ is bound to GCAP1 at EF2, EF3, and EF4, and the structure of each Ca^2+^-bound EF-hand in GCAP1 (Figure 2B) adopts the familiar open conformation as seen in calmodulin and other Ca^2+^-bound EF-hand proteins [57]. Indeed, the interhelical angles for each Ca^2+^-bound EF-hand in GCAP1 are nearly identical to those of GCAP2 (Figure 2A). A unique structural feature of GCAP1 is that the N-terminal α-helix (α1 in Figure 1) and C-terminal helix (α11) are held closely together by their mutual interaction with the N-terminal myristoyl group (Figure 2D). Thus, the covalently attached myristoyl group in GCAP1 is sequestered within a unique environment inside the Ca^2+^-bound protein and prevents GCAP1 from having a Ca^2+^-myristoyl switch [28,56]. In essence, the myristoyl group serves to bridge both the N-terminal and C-terminal ends of the protein, which explains how Ca^2+^-induced conformational changes in the C-terminal domain (particularly in EF4) might be transmitted to a possible target binding site in EF1. A Ca^2+^-myristoyl tug mechanism [58,59] has been proposed to explain how Ca^2+^-induced conformational changes in EF4 serve to “tug” on the adjacent C-terminal helix that connects structurally to the myristoyl group and EF1. This tug mechanism helps explain how Ca^2+^-induced structural changes in EF4 might be relayed to the cyclase binding region in EF1 [60]. The Ca^2+^-induced structural changes involving the C-terminal helix might also be related to Ca^2+^-dependent phosphorylation of S201 in GCAP2 [25].

#### 2.1.3. Ca^2+^-Induced Conformational Changes in GCAP1

The atomic-level structure of Ca^2+^-free/Mg^2+^-bound activator form of wild-type GCAPs is currently not known. A GCAP1 mutant, V77E (called GCAP1^V77E^) was shown previously to abolish dimerization of GCAP1 that significantly sharpened its NMR spectrum, and GCAP1^V77E^ was used to solve the NMR structure of Ca^2+^-free/Mg^2+^-bound GCAP1^V77E^ [48]. The NMR structure of Ca^2+^-free/Mg^2+^-bound GCAP1^V77E^ is shown in Figure 2C. The overall structure of Ca^2+^-free/Mg^2+^-bound GCAP1^V77E^ is similar to the crystal structure of Ca^2+^-bound GCAP1 (root mean squared deviation of main-chain atoms is 2.4 Ǻ when comparing the two structures). The overall structural similarity between Ca^2+^-free and Ca^2+^-bound GCAP1 may explain why GCAP1 has nearly a 100-fold higher Ca^2+^-binding affinity compared to the Ca^2+^ sensor proteins like recoverin and calmodulin that undergo large and unfavorable conformational changes coupled to Ca^2+^ binding [49,57]. In a sense, the GCAP proteins are more like the Ca^2+^ buffer proteins (calbindins and parvalbumin) that adopt pre-formed EF-hand open structures in the absence of Ca^2+^, which allows the buffer proteins to have maximal Ca^2+^ binding affinity [57]. However, small Ca^2+^-dependent structural changes are detected within the EF-hands: Ca^2+^ binding to EF2 reveals a small change in the helix packing angle (Figure 3A). The interhelical angle of the Mg^2+^-bound EF2 (114°, highlighted red in Figure 3A) is slightly more closed than the interhelical angle of Ca^2+^-bound EF2 (110°, highlighted cyan in Figure 3A). A similar Ca^2+^-induced opening of the interhelical angle is also apparent in EF3 (Figure 3B). Thus, the small Ca^2+^-dependent conformational changes in EF2 and EF3 might be functionally important for regulating RetGC. The largest Ca^2+^-induced structural change in GCAP1 is observed in the Ca^2+^ switch helix (residues 169–174 highlighted red in Figure 3C,D). Residues in the Ca^2+^ switch helix (T171 and L174) exhibit Ca^2+^-dependent solvent accessibility. T171 is exposed in the Ca^2+^-free structure, whereas it becomes buried and makes contact with L92 in the Ca^2+^-bound structure. Conversely, L174 is buried and makes contact with L92 in the Ca^2+^-free structure, in contrast to its solvent-exposed environment in the Ca^2+^-bound structure. These Ca^2+^-dependent contacts to the Ca^2+^ switch helix may be important for switching GCAP1 from the Ca^2+^-free activator to the Ca^2+^-bound inhibitor states. The Ca^2+^-induced shortening of the Ca^2+^ switch helix may also serve a role in modulating Ca^2+^-dependent contacts with RetGC.

### 2.2. Dimeric Structures of GCAP1 and GCAP2

The GCAP proteins have a propensity to self-associate as dimers at protein concentrations in the micromolar range or higher [48,60,61,62]. The relatively high dissociation constant of GCAP dimerization may shift into the physiological range if a pre-formed GCAP dimer binds with nanomolar affinity to a RetGC dimer [63] to form a 2:2 complex (GCAP_2_/RetGC_2_) [64]. In essence, the high-affinity binding of GCAP1 to RetGC should shift the apparent dissociation constant of the GCAP1 dimer (bound to RetGC) into the sub-micromolar range. Ca^2+^-induced structural changes to the quaternary structure of a GCAP_2_/RetGC_2_ complex (Figure 4) are proposed here to amplify the relatively small Ca^2+^-induced change in the GCAP1 tertiary structure (Figure 3). Indeed, the binding of Ca^2+^ to GCAP1 was shown previously to cause a 6-fold decrease in the dissociation constant for GCAP1 dimerization [61]. Missense mutations affecting Ca^2+^ binding to GCAP1 also lead to cone-rod dystrophies by altering protein dimerization and functional properties [65]. Thus, Ca^2+^-dependent quaternary structural changes in the GCAP_2_/RetGC_2_ complex may allosterically regulate the RetGC cyclase activity (Figure 4), similar to the allosteric regulation of O_2_ binding to hemoglobin [66]. Recall for hemoglobin, the O_2_-induced change in the tertiary structure of hemoglobin is quite small, but O_2_ binding causes a much larger change in the quaternary structure of the hemoglobin tetramer, known as the T → R transition. A similar allosteric transition may take place in the GCAP_2_/RetGC_2_ complex (Figure 4) and therefore explain how Ca^2+^ binding can modulate cyclase activity with positive cooperativity [8,14]. Atomic-level structures of dimeric forms of GCAP1 [67] and GCAP2 [68] have been reported and were described in a recent review [69]. I will provide an updated overview of the dimeric GCAP structures below.

The atomic-level structure of a GCAP1 dimer (Figure 5A) was modeled previously by a molecular docking approach that used intermolecular distance restraints experimentally measured by EPR-DEER [67] and a separate dimerization model was calculated from small X-ray scattering (SAXS) measurements [61]. The GCAP1 dimer is comprised of mostly hydrophobic intermolecular contacts at the dimer interface (Figure 5B). The most apparent intermolecular contacts involve exposed hydrophobic residues: H19, Y22, M26, V77, and W94 (Figure 5B). A key linchpin contact is formed by the methyl side-chain atoms of V77 that each contact one another at the dimer interface and perhaps explain why the V77E mutation disrupts GCAP1 dimerization [48]. The GCAP1 dimerization site is further stabilized by intermolecular contacts formed by exposed aromatic side chains of H19, Y22, F73, and W94. The point mutation p.H19Y in human GCAP1 that is located in the dimer interface was identified in patients diagnosed with retinitis pigmentosa, and the H19Y GCAP1 mutant protein disrupts RetGC regulation and dimer formation [61,70]. Single point mutations of the hydrophobic residues at the GCAP1 dimer interface (H19A, Y22A, F73A, V77E, and W94A) also each weaken the dimerization dissociation constant and abolish the activation of RetGC by GCAP1 [67]. Thus, the hydrophobic contacts at the GCAP1 dimer interface (Figure 5B) are essential for both its dimerization and activation of RetGCs. This implies that GCAP1 dimerization may be important for activating RetGC and therefore supports the idea of a pre-formed GCAP1 dimer that binds to the dimeric RetGC to stabilize a high affinity 2:2 target complex as discussed above (Figure 4 and Figure 6B). Alternatively, the pre-formed GCAP1 dimer in solution may not exist in the presence of RetGC, because residues in the GCAP1 dimer interface (Figure 5B) appear to overlap with residues that interact with RetGC [71]. Thus, the residues at the GCAP1 dimerization site may prefer to interact with RetGC in the presence of saturating RetGC (Figure 6B), and the binding of RetGC to GCAP1, in this case, would be expected to prevent GCAP1 dimerization. Future studies are needed to probe whether the structure of the GCAP1 dimer (Figure 5A) will remain intact upon its binding to RetGC. In particular, future cryoEM studies are needed to determine the atomic-level structure of RetGC bound to GCAP1.

A structure of the GCAP2 dimer (Figure 5C) was reported previously based on a mass spectrometry analysis [68,72]. The overall quaternary structure of the GCAP2 dimer (Figure 5C) is very different from that of GCAP1 (Figure 5A). In contrast to the dimerization site in GCAP1, the GCAP2 dimerization site is comprised of mostly polar and charged amino acid residues (K98, L167, V171, R175, K183, Q186, D188 highlighted red in Figure 5C). The GCAP2 interface is therefore stabilized primarily by intermolecular salt bridges and hydrogen bonds. The side-chain atoms of R175 in GCAP2 form intermolecular hydrogen bonds with the polar side-chain atoms of Q186 (Figure 5C), and the side-chain atoms of K98 form an intermolecular salt bridge with the side chain carboxylate atoms of D188 (Figure 5C). These intermolecular polar contacts in the GCAP2 dimer are not conserved in the other GCAPs and may explain why the GCAP2 dimer structure (Figure 5C) is quite different from that of GCAP1 (Figure 5A). The different quaternary structures for the GCAP1 and GCAP2 dimers might help to understand their different targeting of RetGC [73,74]. GCAP1 has been shown previously to bind to the kinase homology domain in RetGC [74,75], in contrast to GCAP2 that has been suggested to bind to RetGC residues (Y1016–S1103) at the C-terminus [73].

### 2.3. GCAP5 Is a Fe^2+^ Sensor in Zebrafish Photoreceptors

GCAP homologs are conserved in all vertebrate photoreceptors, and zebrafish photoreceptors contain particular GCAP homologs (GCAP3–5 in Figure 1) [16,76] that do not exist in mammals. The zebrafish homolog called GCAP5 has an amino acid sequence that is perhaps the most divergent of all GCAPs (Figure 1). The first 20 amino acids from the amino-terminus in GCAP5 are particularly unique and contain non-conserved Cys residues (Cys15 and Cys17) that were shown previously to bind Fe^2+^ [62]. One Fe^2+^ binds with nanomolar affinity to two molecules of GCAP5 at the dimer interface and at least two other Fe^2+^ molecules bind to GCAP5 with a dissociation constant in the micromolar range [62]. The nanomolar Fe^2+^ binding to GCAP5 is abolished by the GCAP5 mutations (C15A and C17A), implying that the high-affinity Fe^2+^ is chelated by the sulfhydryl side chains of Cys15 and Cys17. By contrast, the lower affinity Fe^2+^ binding was not affected by the Cys mutations (C15A and C17A). A detailed NMR titration revealed that the lower affinity Fe^2+^ ions are likely binding to the second and third EF-hands in the absence of Ca^2+^ because the micromolar Fe^2+^ binding is abolished in the presence of saturating Ca^2+^ levels. The Ca^2+^-free/Fe^2+^-free/Mg^2+^-bound GCAP5 causes ~10-fold activation of RetGC activity, which is somewhat lower than the cyclase activation promoted by Ca^2+^-free/Mg^2+^-bound GCAP1 [62]. Unlike GCAP1 and GCAP2, both the Ca^2+^-free and Ca^2+^-bound forms of GCAP5 can each activate RetGC. Interestingly, the Fe^2+^-bound GCAP5 is unable to activate RetGC even at low Ca^2+^ levels in light-adapted photoreceptors. The Fe^2+^-induced cyclase inhibition by GCAP5 suggests that Fe^2+^ binding to GCAP5 may serve to modulate cyclase activity and therefore GCAP5 could act as a Fe^2+^ sensor for phototransduction in zebrafish photoreceptors [62].

A structural model of Fe^2+^-bound GCAP5 was determined by an NMR-guided homology modeling approach [62] (Figure 5D). GCAP5 was measured by size-exclusion chromatography to form a protein dimer at micromolar protein concentrations [62] and was accordingly modeled to form a dimer in the structure. The GCAP5 dimer structure (Figure 5D) is somewhat similar to the structure of the GCAP1 dimer (Figure 5A). The GCAP5 dimerization site contains exposed hydrophobic residues (H18, Y21, M25, F72, V76, and W93) that are also present in the GCAP1 dimer (Figure 5B). A single Fe^2+^ is bound to the GCAP5 dimer in which the bound Fe^2+^ is chelated by the side chains of Cys15 and Cys17. The bound Fe^2+^ bridges two GCAP5 molecules into a [Fe(SCys)_4_] dimeric complex [62] like that observed previously in two-iron superoxide reductases [77,78]. The four cysteinyl thiolate groups that ligate the bound Fe^2+^ are similar in structure to the four Cys residues found in the Cys_4_ zinc finger motif that binds to Zn^2+^ [79]. The structural similarity to the Cys_4_ zinc finger suggests that GCAP5 may also bind to Zn^2+^ in place of Fe^2+^. High levels of Zn^2+^ are found in retinal photoreceptor cells, and Zn^2+^ may play a role in phototransduction [80]. Future studies are needed to test whether Zn^2+^ can bind to GCAP5 and test whether Zn^2+^ binding to GCAP5 can regulate RetGCs in zebrafish photoreceptors.

### 2.4. Druggable Hot Spot on the Structure of GCAP1

The structure of GCAP1 reveals exposed hydrophobic residues (H19, Y22, M26, F73, V77, and W94) that are clustered on the surface of the protein and form a potential hot spot for drug targeting (Figure 6A). These exposed hydrophobic residues are located at the GCAP1 dimerization site (Figure 5B), which explains why single mutations to these residues (H19A, Y22A, M26A, F73A, V77E, and W94E) both weaken dimerization and abolish cyclase activation [67]. A schematic model of a preformed GCAP1 dimer bound to RetGC suggests how GCAP1 dimerization might promote cyclase activation (Figure 6B). Alternatively, a monomeric form of GCAP1 bound to RetGC could also promote cyclase activation (Figure 6C), if the exposed hotspot on GCAP1 were to bind directly to RetGC as suggested by [71]. Regardless of whether the exposed hotspot facilitates GCAP1 dimerization (Figure 6B) or binds to RetGC (Figure 6C), this hotspot (highlighted red in Figure 6) could be targeted for drug design. Small molecules or peptides that bind specifically to the hotspot region are expected to block GCAP1 dimerization and/or RetGC binding and should therefore prevent cyclase activation by GCAP1. Small molecule inhibitors that bind to the hotspot region within constitutively active GCAP1 mutants (Y99C, D100G, E111V, and E155G) should block their activation of RetGCs, and therefore diminish the onset of cone-rod dystrophies. Future studies are needed to first screen for drug molecules that bind to the GCAP1 hot spot and then determine whether these drugs can serve as therapeutics for cone-rod dystrophies.

## Figures and Tables

**Figure 1 ijms-22-08731-f001:**
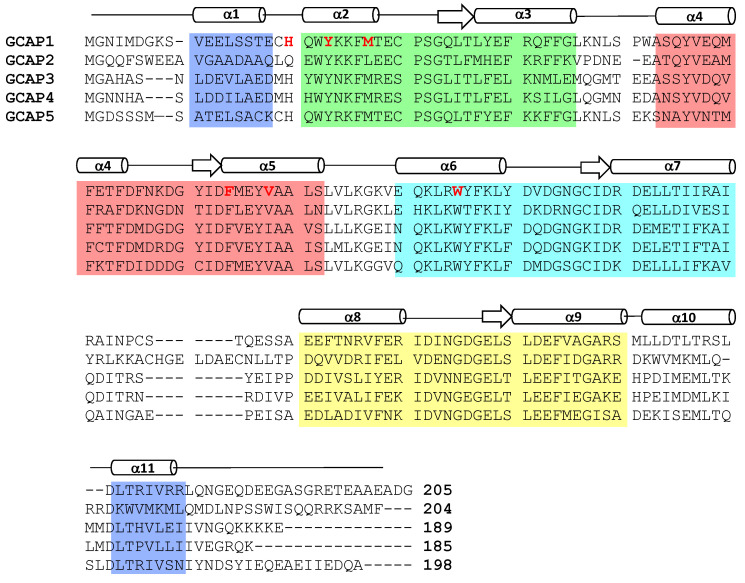
Amino acid sequence alignment of GCAP proteins (bovine GCAP1-2 and zebrafish GCAP3-5). Secondary structure elements (helices and strands) are depicted by cylinders and arrows. EF-hand residues are shaded in green, red, cyan, and yellow. The terminal residues that contact the myristoyl group are shaded purple. Exposed hydrophobic residues at the GCAP1 dimerization site are highlighted in bold and red.

**Figure 2 ijms-22-08731-f002:**
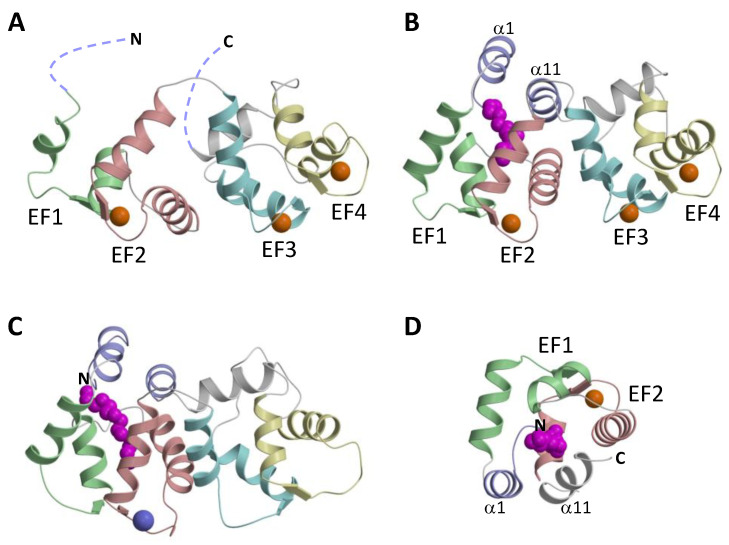
Atomic-level structures of unmyristoylated GCAP2 (**A**), myristoylated GCAP1 (**B**), Mg^2+^-bound GCAP1^V77E^ (**C**), and myristoyl group binding site in Ca^2+^-bound GCAP1 (**D**). The color scheme is the same as in Figure 1. The EF-hands are shaded green, red, cyan, and yellow. The terminal helices (α1 and α11) that contact the myristoyl group are colored purple. Bound Mg^2+^ and Ca^2+^ are colored purple and orange, respectively. The N-terminal myristoyl group is colored magenta.

**Figure 3 ijms-22-08731-f003:**
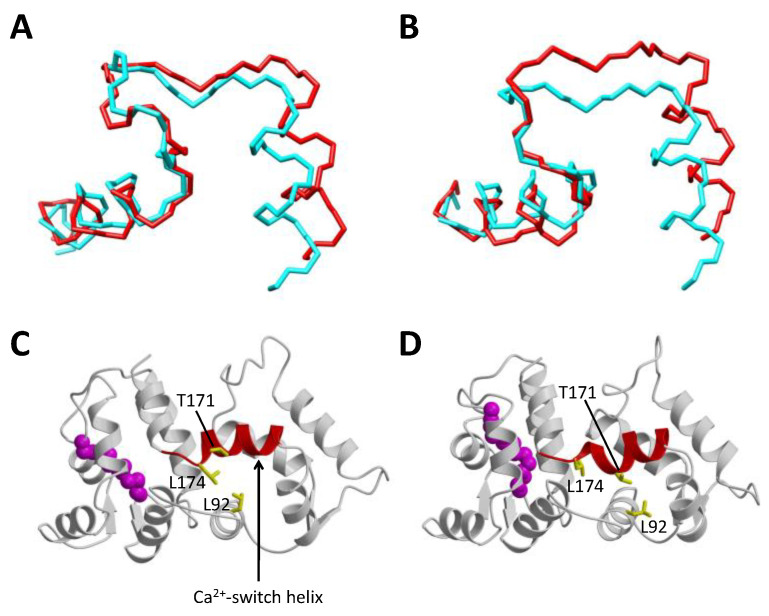
Ca^2+^-induced conformational changes in GCAP1. Main chain structures of EF2 (**A**), EF3 (**B**), Mg^2+^-bound/Ca^2+^-free GCAP1^V77E^ (**C**), and Ca^2+^-bound GCAP1 (**D**). The Ca^2+^-free structures of EF2 and EF3 (red in panels (**A**,**B**)) are overlaid on top of the Ca^2+^-bound structures (cyan). EF2 and EF3 exhibit a Ca^2+^-induced decrease in interhelical angle. The Ca^2+^-switch helix (residues 169–174) undergoes a Ca^2+^-induced shortening (highlighted red in panels (**C**,**D**)).

**Figure 4 ijms-22-08731-f004:**
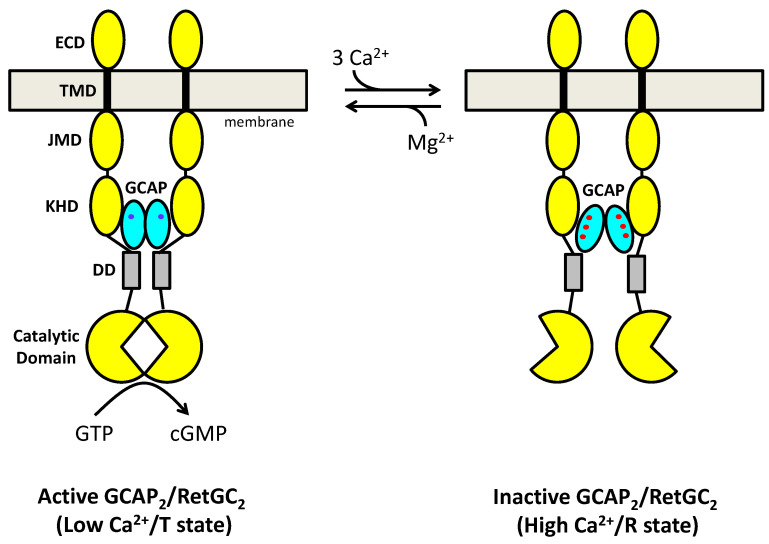
Allosteric regulation of a GCAP_2_/RetGC_2_ complex. Cyclase activity (synthesis of cGMP) is modulated by a Ca^2+^-dependent change in the quaternary structure of the GCAP_2_/RetGC_2_ complex. The Ca^2+^-free/Mg^2+^-bound GCAP1 dimer (cyan ovals with bound Mg^2+^ in blue) binds to the RetGC dimer (yellow) and activates cyclase activity (left panel). The Ca^2+^- bound GCAP1 dimer (cyan ovals with three bound Ca^2+^ in red) binds to the RetGC dimer (yellow) and inactivates cyclase activity (right panel). Thus, the binding of 3 Ca^2+^ to the GCAPs promotes the T → R transition (turns off cyclase activity), whereas the dissociation of Ca^2+^ and binding of Mg^2+^ promotes the R → T transition (turns on cyclase activity). Each RetGC dimer subunit is composed of an extracellular domain (ECD), transmembrane domain (TMD in black), juxtamembrane domain (JMD), kinase homology domain (KHD), dimerization domain (DD, gray), and catalytic cyclase domain (notched circles).

**Figure 5 ijms-22-08731-f005:**
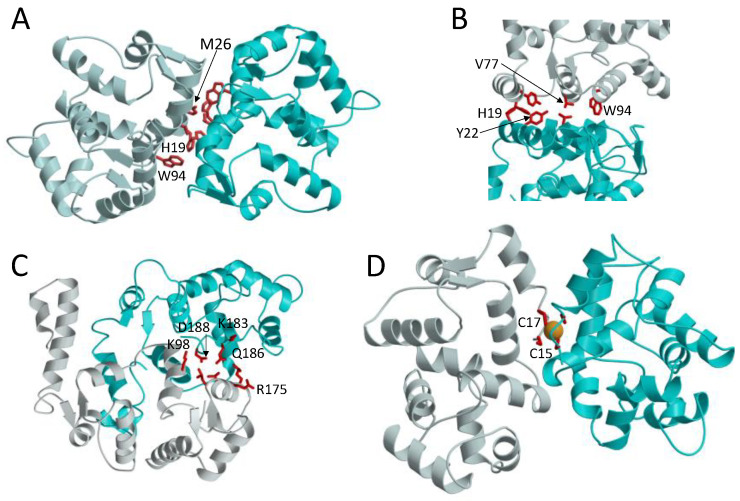
Dimeric structures of GCAP1 (**A**), GCAP2 (**C**), and GCAP5 (**D**). A close-up view of the GCAP1 dimerization site (**B**) reveals intermolecular contacts between aromatic residues (red). The GCAP2 dimerization site is stabilized by intermolecular salt bridges and hydrogen bonds (highlighted by red residues in panel (**C**)). The GCAP5 dimerization site is stabilized by a bound Fe^2+^ (orange sphere) that is chelated by C15 and C17 (**D**).

**Figure 6 ijms-22-08731-f006:**
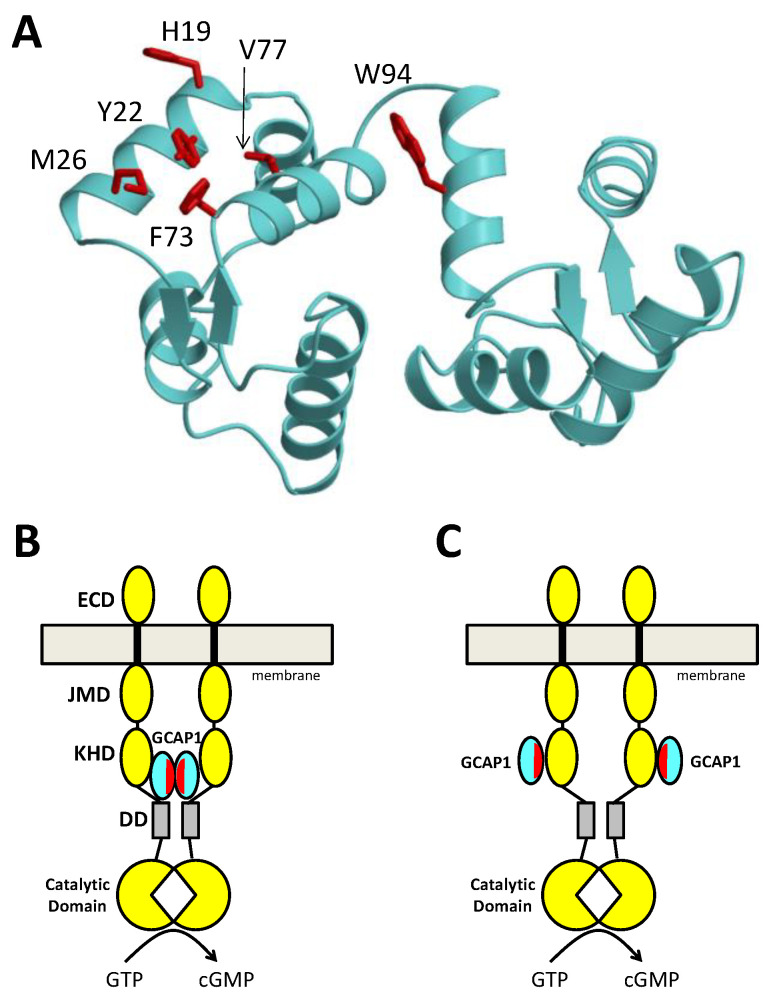
Druggable Hotspot on GCAP1 (**A**) and RetGC activation by dimeric (**B**) or monomeric (**C**) GCAPs. The structure of GCAP1 (cyan) contains exposed hotspot residues (red) that can mediate GCAP dimerization (**B**) or RetGC binding (**C**). RetGC (yellow) is proposed here to be activated by either a preformed GCAP1 dimer (**B**) or by monomeric GCAP1 (**C**). Small molecule drugs or peptides that bind to the hotspot are expected to prevent cyclase activation by constitutively active GCAP1 mutants and therefore may serve as therapeutics for cone-rod dystrophies. Each RetGC dimer subunit (yellow) is composed of an extracellular domain (ECD), transmembrane domain (black), juxtamembrane domain (JMD), kinase homology domain (KHD), dimerization domain (gray), and catalytic cyclase domain.

## Data Availability

Not Applicable.

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
