# Peer review of "Structural Insights into Retinal Guanylate Cyclase Activator Proteins (GCAPs)"

_ijms, 2021, doi:10.3390/ijms22168731_

Round 1
Reviewer 1 Report
This review article describes structure of GCAPs and potential mechanism of Ca2+ dependent activation of retGC by GCAP. The article also discusses plausible mechanisms of disease causing mutations. The article also touches briefly potential pharmacological intervention of GCAP function based on structural knowledge. The article is brief but well written and good introduction to the field. Each section has an informative descriptive title. The headings (1. Introduction and 2. Results and Discussion) do not match the contents. They should be removed and new numbers should be given to each section. The article will be improved if short introductory remark and concluding remark are added. The list of the references does not conform to the journal format and need to be fixed. Here are more specific comments.
L151-L182 The discussion is much easier to visualized if Fig. 3 shows amino acid residues discussed.
L191-L192 the dissociation constants should be listed to better describe the shift.
L254 the reference should be numbered and included in the reference list.
L264 It is not clear. Targeting difference between GCAP1and GPAC2 should be discussed in more detail.
Author Response
Responses to the reviewer comments are in italics below:
***** (ijms-1329430) *****
Reviewer(s)' Comments to Author:
Comment:
L151-L182: The discussion is much easier to visualized if Fig. 3 shows amino acid residues discussed.
Response:
The side chains of amino acid residues (L92, T171 and L174) have now been added to Fig. 3 as requested.
Comment:
L254: the reference should be numbered and included in the reference list.
Response:
This reference is now numbered as 72 and is included in the reference list.
Comment:
L264: It is not clear. Targeting difference between GCAP1 and GPAC2 should be discussed in more detail.
Response:
A sentence has been added at lines 269-272 that explains in more detail the different binding sites in RetGC that have been proposed for GCAP1 and GCAP2.

Reviewer 2 Report
The review by James Ames describes what is known about structure and functions of the GCAP proteins. It is an agile and quite precise summary with interesting data about the last experimental evidences in the literature including the dimeric form of GCAP1 and the iron binding capability of zebrafish GCAP5. Moreover it proposes an interesting model about the possible involvement of GCAP dimeric assembly in controlling the retGC activity.
I have only few suggestions:
It is worth mentioning the different propensity to form dimeric assemblies when GCAP1 is bound to Ca or Mg that could be related to the protein function, reporting the experimental Kd of GCAP1 dimer (as in reference 60 in the manuscript).
Since a higher dimeric propensity of GCAP1 is related to its inhibitory activity I am wondering if the presence of iron in GCAP5 can induce the stabilization of the dimeric assembly and enhance its inhibitory activity.
There is a missing parenthesis at row 71
Row 144: a reference is missing: I suggest to add ‘Allosteric communication pathways routed by Ca 2+/Mg 2+ exchange in GCAP1 selectively switch target regulation modes’ V Marino, D Dell’Orco Scientific reports 6 (1), 1-14.
Row 153-154: the sentence is not correct since both Ca or Mg bound forms of GCAP1 tend to form dimers albeit with different propensity (higher for the calcium bound form)
In paragraph 2.2 the author should mention the existence of other different models for the dimeric assembly of human GCAP1 in the presence of Ca obtained by SAXS data (as described in reference 60).
In paragraph 2.2 the author should mention that some diseases associated point mutations are shown to perturb the quaternary assembly of the protein (analyzed by SAXS) again suggesting a possible correlation between dimerization and regulation (suggested ref.: 'Missense mutations affecting Ca2+-coordination in GCAP1 lead to cone-rod dystrophies by altering protein structural and functional properties'. Dal Cortivo G, Marino V, Bonì F, Milani M, Dell'Orco D. Biochim Biophys Acta Mol Cell Res. 2020 Oct;1867(10):118794)
Author Response
Responses to the reviewer comments are in italics below:
***** (ijms-1329430) *****
Reviewer(s)' Comments to Author:
Comment:
It is worth mentioning the different propensity to form dimeric assemblies when GCAP1 is bound to Ca or Mg that could be related to the protein function, reporting the experimental Kd of GCAP1 dimer (as in reference 60 in the manuscript).
Response:
A sentence has now been added at lines 197-199 indicating that Ca2+ binding to GCAP1 causes a 6-fold decrease in the dissociation constant for GCAP1 dimerization as reported by ref 60.
Comment:
Since a higher dimeric propensity of GCAP1 is related to its inhibitory activity I am wondering if the presence of iron in GCAP5 can induce the stabilization of the dimeric assembly and enhance its inhibitory activity?
Response:
GCAP5 was previously shown by Lim et al (2017) to form a dimer in both the presence and absence of Fe2+. So, Fe2+ binding to GCAP5 is not required for dimerization. However, the effect of Fe2+ binding on the dimer dissociation constant is currently not known.
Comment:
There is a missing parenthesis at row 71
Response:
The missing parenthesis is now added.
Comment:
Row 144: a reference is missing: I suggest to add ‘Allosteric communication pathways routed by Ca 2+/Mg 2+ exchange in GCAP1 selectively switch target regulation modes’ V Marino, D Dell’Orco Scientific reports 6 (1), 1-14.
Response:
This reference (reference number 59) has been added.
Comment:
Row 153-154: the sentence is not correct since both Ca or Mg bound forms of GCAP1 tend to form dimers albeit with different propensity (higher for the calcium bound form).
Response:
This sentence has been removed.
Comment:
In paragraph 2.2 the author should mention the existence of other different models for the dimeric assembly of human GCAP1 in the presence of Ca obtained by SAXS data (as described in reference 60).
Response:
A sentence has been added at lines 224-225 indicating that a separate dimerization model was calculated from small xray scattering (SAXS) measurements.
Comment:
In paragraph 2.2 the author should mention that some diseases associated point mutations are shown to perturb the quaternary assembly of the protein (analyzed by SAXS) again suggesting a possible correlation between dimerization and regulation (suggested ref.: 'Missense mutations affecting Ca2+-coordination in GCAP1 lead to cone-rod dystrophies by altering protein structural and functional properties'. Dal Cortivo G, Marino V, Bonì F, Milani M, Dell'Orco D. Biochim Biophys Acta Mol Cell Res. 2020 Oct;1867(10):118794)
Response:
A sentence has been added to lines 199-200 indicating missense mutations affecting Ca2+ binding to GCAP1 lead to cone-rod dystrophies by altering protein dimerization and functional properties as reported by Dal Cortivo et al 2020.
